# Protein X-ray Crystallography and Drug Discovery

**DOI:** 10.3390/molecules25051030

**Published:** 2020-02-25

**Authors:** Laurent Maveyraud, Lionel Mourey

**Affiliations:** Institut de Pharmacologie et de Biologie Structurale, IPBS, Université de Toulouse, CNRS, UPS, 31400 Toulouse, France

**Keywords:** X-ray crystallography, drug discovery, protein-ligand complexes, structure-based drug design, ligand screening, therapeutic targets, three-dimensional structures, high resolution

## Abstract

With the advent of structural biology in the drug discovery process, medicinal chemists gained the opportunity to use detailed structural information in order to progress screening hits into leads or drug candidates. X-ray crystallography has proven to be an invaluable tool in this respect, as it is able to provide exquisitely comprehensive structural information about the interaction of a ligand with a pharmacological target. As fragment-based drug discovery emerged in the recent years, X-ray crystallography has also become a powerful screening technology, able to provide structural information on complexes involving low-molecular weight compounds, despite weak binding affinities. Given the low numbers of compounds needed in a fragment library, compared to the hundreds of thousand usually present in drug-like compound libraries, it now becomes feasible to screen a whole fragment library using X-ray crystallography, providing a wealth of structural details that will fuel the fragment to drug process. Here, we review theoretical and practical aspects as well as the pros and cons of using X-ray crystallography in the drug discovery process.

## 1. Introduction

The first X-ray diffraction by protein crystals was reported in the early thirties [1,2], but nearly 30 years passed before the atomic crystallographic structure of myoglobin was published [3]. Yet, the potential of X-ray crystallography was already evidenced as it allowed the unambiguous structure determination of penicillin [4]. X-ray diffraction, either on crystalline material or on amorphous powders, is still routinely used in the pharmaceutical industry for the characterization of drugs, drug polymorphism, and pseudopolymorphism [5,6]. The idea that the knowledge of a protein structure could help the design of specific ligands, which is now a widely accepted obviousness, appeared in 1976, a few years after the launch of the Protein Data Bank, in 1971 [7,8]. It then grew in importance to the point that the “rational drug design cycle” was elaborated [9]: a drug can be rationally designed and optimized using the knowledge provided by the structure determination of its complex with the macromolecular target, which will guide the medicinal chemist for drug optimization. The interaction of the optimized compounds with the drug target is structurally characterized, allowing the next cycle of chemical optimization. The core of this cycle still applies nowadays, although with more sophistication (Figure 1). With the advent of crystallization automates, brighter synchrotron X-ray sources, faster detectors, automated structure solution, and refinement pipelines, it is not uncommon to be able to determine macromolecular structures within a few days [10], and in the case of structures of protein-ligand complexes, throughput of several structures a day is attainable [11]. As the throughput augmented, the use of X-ray crystallography progressed from the protein target structure determination, possibly in the presence of some pre-identified ligand, to structure-activity relationship determination by X-ray crystallography, where several structures of complexes are determined in order to guide ligand optimization [12], to a screening tool of chemical libraries containing a few hundreds of compounds distributed as cocktails of up to ten compounds [13] or even individually as it is now feasible [14].

## 2. Theoretical Aspects

As X-rays interact with the electrons of atoms, X-ray crystallography provides an electron density map, which localizes electrons in the asymmetric unit of the crystal. This electron density map is a time- and space-average of the electron density of molecules present in the crystal; part of molecules that are highly flexible will therefore be associated with poor electron density and will usually be missing in the final model. Similarly, molecules that are present only in a fraction of the unit cells of the crystals, such as low molecular weight ligands, will display a weak electron density, which will overlap with the electron density of surrounding, and potentially ordered, solvent molecules. The refined model derived from the interpretation of the electron density map, as available from the Protein Data Bank [15], will include, for each atom of the model, the atom type, the Cartesian coordinates, the occupancy, and the atomic displacement parameter (ADP). It should be kept in mind that the atom type, i.e., the element, is usually derived from prior knowledge of the chemical entity present in the crystal. Indeed, X-ray crystallography usually does not allow to distinguish between C, N, and O, and sometimes even S, as they have a similar number of electrons. In addition, hydrogen atoms, with their lone electron, are hardly visible in electron density maps. Coordinates of an atom indicate its position, whereas the occupancy, which is comprised between 0.0 and 1.0, is the measure of the fraction in which the atom occupies this position. This occupancy can be seen as the fraction of time the atom spends at this position (dynamics disorder), the fraction of the unit cell in the crystals where the atom is actually found at this position (static/statistic disorder), or a combination of both. The ADP can be isotropic, and hence described with a single number called the isotropic B-factor in Å^2^, which is usually the case for structure at a resolution lower than about 1.5 Å, or anisotropic, which will require six numbers, in case high-resolution data (i.e., better than 1.5 Å) are available. This ADP, which actually describes the attenuation of X-ray scattering as the result of both thermal motion and static disorder [16], is proportional to the square of the mean atomic displacement, and can be seen as the spherical, respectively ellipsoidal, region that the atom explores centered on its Cartesian coordinates, in an isotropic, respectively anisotropic, case. It is beyond our scope to provide a detailed description of macromolecular crystallography, as many reviews and textbooks are available (see for example [17]). The following paragraph nevertheless briefly highlights some fundamental principles for determining crystallographic structures of biological macromolecules.

The interaction of a monochromatic X-ray beam with an ordered and periodic molecular object, such as a three-dimensional crystal, will generate a diffraction pattern where X-ray waves are diffracted in definite directions with definite amplitudes and phases. The directions of these diffracted waves, or the positions of the diffracted spots on the detector, depend on the three vectors that define the periodicity of the crystal, which in turn defines the unit cell of the crystal. The amplitudes and phases reflect the molecular structure of the crystallized (macro)molecule. If these are known, then an electron density map can be computed and the corresponding structure can be determined. The data collection experiment allows for the measurement of diffracted beam intensities, which are proportional to the square of the diffracted wave amplitudes. The phases of these waves are not measurable experimentally and specific phasing methods are required to recover this information. Among these methods, molecular replacement will usually provide phase information if a homologous structure is already available in the Protein Data Bank. If no homologous structure is available or if the molecular replacement fails, experimental phasing is needed. The latter will provide unbiased phase information, but will require additional, and sometimes tedious, experiments. Experimental phasing usually relies on the isomorphous introduction of a specific atom (with a high number of electrons and/or with anomalous scattering properties, such as mercury or selenium) into the protein crystal and on the collection of additional diffraction data from these derivatized crystals. In some favorable cases, the weak anomalous signal of sulfur atoms, natively present in cysteine and methionine side-chains, can provide enough information for solving the structure in a S-single wavelength anomalous dispersion (S-SAD) experiment [18]. With both methods, the obtained phases are combined to the observed amplitudes, allowing for the computation of an experimental electron density map using inverse Fourier transform. An initial atomic model is then built in this electron density map and refined. Alternation of model modification and completion with computational refinement eventually results in the best possible model given the experimental data. This will not only include protein atoms but also ordered solvent molecules like water or potentially any compound present in the crystallization solution. Fitting atoms in an electron density map and building an accurate and reliable model of the investigated molecule is a complex and multi-parameter task. Although many tools are available for evaluating the reliability of a protein structure, based on the analysis of quality of the stereochemistry of the model as well as on the agreement with experimental data [19], quality of the refinement is often summarized with two numbers, the so-called *R* and *R_free_* factors. These values correspond to the ability of the model to explain experimental data, for data that were used for building and refining the model, in the case of the *R-*factor, or for a subset of the data that was not used to derive the model, in the case of the *R_free_* [20]. An important contribution of X-ray crystallography is that it provides at rather “low cost” a precise picture of the protein hydration in the crystal form. Water molecules associated with crystallized proteins are normally of two types consisting of what was termed “structural waters”, which correspond to individual, discrete, firmly bound molecules, and more loosely ordered surrounding shells. Scrutinizing water molecules, or any other solvent molecules, might reveal useful to drug designers when building drugs [21].

## 3. Workflow

The classical workflow of a rational drug design project (Figure 1) is usually initiated with the structure determination of the—supposedly validated—pharmacological target. X-ray crystallography is the more common tool for this endeavor, but nuclear magnetic resonance (NMR) and, more recently, cryo-electron microscopy (cryo-EM) are also important players in the field. In the absence of experimental structural information, modelling of the target’s structure can also provide useful insights [22,23], at the condition that at least one structure of a homologous protein exists at the Protein Data Bank (PDB) [24]. Beside the structure of the pharmacological target, at least one ligand has to be identified in order to initiate the rational drug design process. Ligand identification can be addressed using several technologies:
High-throughput screening (HTS) allows one to experimentally screen large chemical libraries, if an appropriate assay is available, resulting in a collection of potential ligands. Preliminary confirmation of direct binding to the protein target using biophysical methods will help to select the most appropriate compounds for structural studies.It is quite common that in the course of the characterization of a protein target, some knowledge is gained on potential ligands, such as substrate or cofactor analogues, which are often potent inhibitors.Success in target structure determination opens access to virtual screening methods, which aim at identifying potential ligands of a target using computational methods. This step serves often as a preliminary filter to identify a few hundred potential binders among huge virtual libraries, in order to reduce the experimental work needed to confirm the interaction. Experimentally confirmed ligands are then used in structural studies.Recently, fragment-based drug discovery (FBDD), also known as fragment-based drug design, and fragment-based ligand/lead discovery (FBLD) has emerged as a powerful tool to identify ligands, albeit with poor affinity given the reduced molecular weight of fragments (about 150–250 Da, containing 10 to 20 non-hydrogen atoms). With such compounds, detection of binding requires sensitive techniques, and X-ray crystallography has proved to be very effective at identifying weak binders, with affinity as low as a few mM. In most favorable cases, fragment libraries of a few hundreds of compounds can be screened using X-ray crystallography, delivering dozens of structures of complexes [11,25]. Although FBDD was initially used as an alternative approach to HTS, i.e., for targets without hits, it appears that these two methodologies are more and more often used in parallel. Despite numerous fragment-based screening technologies existing, X-ray crystallography and protein-based NMR are getting more and more popular since they provide direct and experimental structural information on fragment mode of binding. More recently, the concept of fragment-based screening has been extended to crystallographic screening of ultra-low-molecular weights compounds, called MiniFrags and typically containing 5 to 7 non-hydrogen atoms. Given the low affinity of these compounds, this new crystal-soaking methodology requires working at very high concentrations, typically 1 M [26]. As of October 2018, more than 40 FBLD-derived drugs have been in clinical development, three of them having been approved for clinical use (Table 1).

Knowledge of the structure of a target with a first bound compound, the hit, provides structural information allowing the medicinal chemist to design chemical modifications susceptible to improve the interaction with the target, and therefore the potency. However, improving the interaction is not the sole consideration that should guide the so-called “hit2lead” process, as ADME-Tox properties should also be improved in order for the final drug to be used in humans. Moreover, as efficient binding to the target in vitro does not necessarily translate into an efficient drug in vivo, biological evaluation of drug potency has to be introduced in the process as early as possible. Multiple chemical evolutions of the hit(s) are usually undertaken, and structure activity relationships (SAR) have to be deciphered. This hit2lead process can therefore be seen as an iterative process where each cycle delivers improved compounds (Figure 1). Structure determination at each cycle of the process is of crucial importance for the chemist in order to validate the expected binding mode of the modified compound and helps to rationalize SAR.

## 4. Practical Aspects

Structure determination of a complex formed of a protein and a small molecular weight compound does not present any specific difficulties from the crystallographic point of view, especially if the structure of the protein is already known. However, in the case of a crystallographic fragment screening campaign, the process is far more complicated because of the amount of data that is generated. Indeed, hundreds of crystals are to be prepared hopefully resulting in as many datasets and structures to be solved, refined, and analyzed. With such an amount of data to be processed and analyzed within a short time, automation is of paramount importance, both for crystal preparation and handling, and for computing tasks. Besides, a properly dimensioned computing environment should be available, as terabytes of data will have to be processed.

### 4.1. Crystallizing the Target

Whether engaged in a crystallographic screening or in a hit2lead process, many crystallographic structures of complexes will have to be obtained. It is therefore useful to carefully optimize the crystallization procedure of the target to ensure that it reproducibly delivers a suitable crystal form diffracting to the highest possible resolution, preferentially better than 2.5 Å. In some extreme cases, this step will require to optimize the protein construct and/or the expression and purification steps [27]. Seeding, that is the introduction of microcrystals or crystal fragments into an equilibrated crystallization drop, might be considered in order to attain suitable reproducibility [28,29]. Crystallization conditions allowing for cooling of the crystal to 100 K without the need of prior cryoprotection will definitely facilitate the crystal harvesting steps. The targeted binding site of the protein should not be blocked by crystal contacts with symmetry mates, nor should it be occupied by a molecule coming from the crystallization solution since such compound could compete with ligand binding. Tolerance to the presence of small amounts of the organic solvent used to solubilize putative ligands to be studied should be ensured. From the crystallographic point of view, in order to facilitate data collection, a crystal form with a high symmetry space group should be preferred, i.e., triclinic and monoclinic space groups would necessitate more data to be collected.

### 4.2. Preparation of Crystals of Complexes

#### 4.2.1. Soaking

Soaking experiments are easy to set up and require only small amounts of ligand. A single crystal is immersed in a 1 µL droplet of a 10–100 mM solution of ligand, for a period of time ranging from a few seconds to several days. Soaking is well suited when multiple ligands are planned to be investigated, since the production of protein crystals is well controlled and does not have to be optimized for each assay. The solubility of the ligand is often a limiting factor. Despite the fact that it is difficult to determine the ligand concentration, adding the ligand in the solid form or from saturated solution directly to the crystal may be successful [30]. For soluble ligands, their concentration should be such that 90% of the protein is bound to the ligand, meaning that a ligand concentration at least ten times the *K*_d_ value is required. Because the binding affinity is often not known, ligands are used at concentrations ranging from 0.1 to 1 mM and from 20 to 50 mM for high-molecular-weight ligands and fragments, respectively. Soaking may allow/accommodate small conformation changes but co-crystallization should be preferred in case large changes are expected upon ligand binding. It is not uncommon that soaking experiments render the crystal fragile. In that case, cross-linking of the crystal can be attempted prior to soaking [31]. Although this method often results in the apparition of cracks on the crystals, detrimental for diffraction [32], it has proved to be instrumental in several cases [33]. Soaking crystals when many compounds are to be evaluated might be a tedious process, as the addition of the concentrated ligand solution on the crystal or the transfer of each crystal in a concentrated droplet of ligand is manually performed. However, automation of the soaking procedure has recently been developed using for instance acoustic dispensing of nanodrops of concentrated ligands into crystal containing droplets, where several hundreds of soaking experiments can be performed within minutes [34].

#### 4.2.2. Co-crystallization

For co-crystallization experiments, the protein has to be conditioned in the presence of the ligand prior to crystallization. As the presence of the ligand itself might alter the crystallization conditions, screening of new crystallization conditions might be necessary, in the worst case for each ligand. Pre-coating of crystallization plates with ligands prior to crystallization, followed or not by in situ X-ray diffraction, provides a very practical approach elegantly adapted for fragment screening in 96-well format [36,37]. As for soaking experiments, the ligand concentration that should be used depends on the binding affinity and should be several times greater than the *K*_d_. In the case of ligands with poor solubility, incubation of diluted protein and ligand followed by concentration may be necessary. Temperature and time of incubation prior to launching crystallization assays may vary from 4 °C to room temperature and from 1 h to overnight, respectively.

### 4.3. Crystal Mounting and Data Collection

X-ray diffraction data collection, whether at a synchrotron or on a home source, usually requires to harvest crystals from the crystallization drop, using a loop or mesh, and to cool them at 100 K in order to limit radiation damages [38]. If performed manually, these steps are quite time consuming, especially if an additional cryoprotection step is necessary. Various automation procedures have been developed [39,40]. Some synchrotron beamlines now offer automated sample loading and centering coupled to unattended data collection and processing [41]. An average throughput of 120 crystals/day has been reported at the MASSIF-1 beamline at the European Synchrotron Radiation Facility (ESRF) [42], which includes crystal mounting, centering, data collection, and data processing with multiple pipelines. An elegant alternative to this data collection strategy is the in-situ data collection, where diffraction data are collected at room temperature directly from crystallization drops in 96-well plates [36]. When coupled to the use of precoated plates with dried ligands, this method eliminates the need of crystal harvesting and cooling. However, high quality crystals are needed as X-ray scattering from the crystal is contaminated with the scattering by the plate and the crystallization medium. Additionally, a complete rotation of the crystal in the X-ray beam is not possible. The available rotation range is restricted to about 90°, as the 96-well plate has to remain more or less perpendicular to the beam. Combination of multiple datasets are therefore commonly required, and high-symmetry space groups are privileged. The use of Mylar-based film sandwich plates has recently been described that allows room temperature and cryogenic high-throughput in situ data collection [43].

### 4.4. Data Processing, Structure Determination, and Refinement

In the hit2lead process, structures of complexes allow one to guide the next chemical optimization cycle. Hence, at each cycle, a few structures will be solved and low- to medium-throughput is expected. The use of automated data processing and refinement pipelines will be helpful, but standard, manual crystallographic procedures are still applicable. In contrast, in a crystallographic fragment screening project, where several hundreds of structures need to be determined and analyzed within a few hours or days, automation is mandatory.

The data processing step corresponds to the analysis of X-ray diffraction images for the extraction of diffraction spots intensities. The derived observed structure factors are the actual experimental data that will allow structure determination and refinement. Most synchrotron beamlines offer automatic data processing using various widely spread pipelines, such as AutoProc [44], XDSAPP [45,46], xia2 [47], or locally developed ones, such as GRENADES at ESRF [48] or XDSme at SOLEIL (Legrand P. (2017), GitHub repository https://github.com/legrandp/xdsme ‘XDSME: XDS Made Easier’, Synchrotron SOLEIL, Gif-sur-Yvette, France). These pipelines rely on the use of one of the standard data processing software: XDS [49], iMosflm [50], or Dials [51]. Each software uses a different assumption and algorithm, and the best processed dataset has to be selected for structure determination and refinement. Structure determination is usually straightforward as the structure of the target is available, and the crystallization procedure has been optimized beforehand to reproducibly deliver a selected crystal form. In some cases, when the presence of the ligand induces large cell parameter modifications, or even space group changes, molecular replacement might be needed, but usually a few cycles of rigid body refinement suffice.

In a screening project, full refinement of all generated structures is not desirable since only a few structures will actually display a ligand. The objective is to refine sufficiently the structure to the point that the absence of the ligand is reasonably certain, so that the refinement can be abandoned. The efforts can therefore be concentrated on the refinement of the few actual complexes. In a fragment-screening project, a hit rate of 5–10% is expected [11,52,53,54]. This corresponds to 50 to 100 structures that should be fully refined for a 1000-fragment library. Full refinement of non-bound structures is time consuming and will produce no usable information, and should be avoided as much as possible. Automatic refinement pipelines include the phenix.ligand_pipeline [55], dimple from the CCP4 package [56], and Buster/TNT [57]. It has been suggested that up to a quarter of the ligands could remain undetected if refinement is not pushed to full completion [58]. Indeed, presence of the ligand is easily confirmed for nearly fully occupied tightly bound compounds, as clear positive peaks in the Fourier difference maps will pop up even at an early stage of refinement. On the contrary, low occupancy fragments remain easily overlooked unless specific detection procedures are used (Figure 2). It should be stressed that the interpretation of the electron density maps is often performed by a human. In an extreme case, careful examination of the electron density maps resulted in the retraction of the original structure of the complex [59]. Less dramatic errors can result in bad positioning of part of the ligand, especially as X-ray crystallography does not allow for the distinguishing of carbon, nitrogen, and oxygen atoms in the electron density maps. However, such distinction may confidently rely on the interpretation of the possible hydrogen bond network with surrounding protein atoms or solvent molecules [60,61]. Different procedures have been described in order to detect those weakly-bound ligands. The Pipedream pipeline (Sharff A, Keller P, Vonrhein C, Smart O, Womack T, Flensburg C, Paciorek C and Bricogne G (2011). Pipedream, version 1.2.3, Global Phasing Ltd., Cambridge, United Kingdom) analyzes individual structures independently and treats the expected ligand-binding site specifically, for instance with respect to bulk solvent correction, in order not to obscure the corresponding electron density. A more global approach has been elaborated in the PanDDA procedure [62]: an average electron density map is computed from multiple isomorphous apo-structures, after proper superimposition, and significant deviations from the mean in individual datasets are flagged as events. Each structure of complexes is then refined as an ensemble of bound and unbound states [63]. This procedure is able to identify weakly bound compounds, when the classical Fourier difference analysis fails to detect any (Figure 2).

Refining structures of complexes that include low molecular weight compounds is not straightforward as these compounds are usually not recognized by refinement programs. Thus, a proper dictionary has to be elaborated in order to describe the ideal stereochemical parameters of any “new” molecule. Most refinement programs include dedicated tools that are usually suited for case-by-case generation of suitable restraints, at least in their basic academic version (Grade for BUSTER/TNT, libcheck for CCP4, REEL for PHENIX). Dedicated servers are also available, such as PRODRG2, which are able to generate proper restraints for various refinement programs [64]. These tools may present some limitations in the type of atoms they can properly handle, e.g., metals, or in the way they can automatically process specific chemical groups, e.g., ionizable moieties. Most of these tasks are easily handled when only a few structures have to be determined. However, when many structures have to be handled, like in a crystallographic fragment screening project, dedicated data management and/or workflow tools are needed. XChemExplorer [65] was specifically designed to handle crystallographic screening projects from the diffraction data processing step up to the identification of complexes with PanDDA and deposition of refined structures within the PDB. Although XChemExplorer has been tailored to manage data collected at the diamond light source, it is possible to adapt it to data collected at any source. This is not the first platform designed to handle such cases: this task may also be performed using the AutoSolve platform from the proprietary program Astex [66].

## 5. Potential Pitfalls

The first and most obvious source of failure in generating crystals/structures of complexes is directly linked to the high attrition rate in early experimental steps of a protein structure determination, namely production-purification and crystallization. In that respect, bioinformatics tools might prove useful to facilitate prioritization of the protein targets, find surrogates, suggest the design of protein constructs, and even predict crystallization propensities based on protein sequences [67]. Even though crystals of the selected target have been obtained, it might still reveal difficult to generate conditions for obtaining crystals/structures of the desired complexes. The main culprits for this may be due to differences with respect to the conditions used in the original screening assay (e.g., using different protein constructs: truncated forms, post-translational modification heterogeneities, etc.) and/or changes in crystallization conditions linked to changes in physicochemical conditions, which in turn could lead to decreased affinity of the ligand [27]. Mixture of free protein and protein in complex in the crystal may also result in poor electron density and hence impede fitting the ligand in the structure. In that respect, whichever method is used to obtain co-crystals, i.e., co-crystallization or soaking, ligand solubility is more than often an issue. Indeed, ligands are rarely soluble in the crystallization or stabilization solution rendering necessary the use of an organic solvent. Care should therefore be taken to ensure that the selected solvent, most often dimethyl sulfoxide (DMSO), at the concentration required for delivering a sufficient amount of compound, is compatible with the native state of the protein. In some rare cases, differences were observed between structures of complexes obtained through soaking or co-crystallization [68]. As soaking implies the use of preformed crystals of the protein target, any conformational adaptation to ligand binding might be restricted and the obtained structure might not reflect the binding modes that would be observed in the solution. It is also important to supplement the cryoprotection solution with the ligand to avoid soaking the ligand back. On the other hand, back-soaking in the stabilizing and/or cryoprotection solution may be useful to soak out any ligand previously bound to the protein. Finally, failure may also arise because of false positives obtained during a screening campaign, which strengthens the use of orthogonal methods to confirm an initial hit and to validate ligand binding, using biophysical methods (reviewed in [69]) prior to crystallographic analysis of putative complexes.

## 6. Examples

Captopril, an antihypertensive angiotensin-converting enzyme inhibitor, was the first drug derived from a structural model of a protein. Here, the model of angiotensin-converting enzyme was based on the well structurally characterized zinc metallopeptidase carboxypeptidase A [70]. The first approved drug obtained by means of structure-based drug discovery was dorzolamide, launched by Merck as Trusopt, a topical carbonic anhydrase inhibitor used to treat ocular hypertension and/or glaucoma [71]. The impact of crystallography to support drug discovery is also illustrated by the case of the Bcr-Abl oncoprotein. Bcr-Abl possesses a constitutively activated Abl tyrosine kinase domain leading to unregulated phosphorylation of proteins in hematological stem cells that results in chronic myelogenous leukemia (CML). Imatinib (aka Gleevec, Novartis Pharma AG), identified as a Bcr-Abl inhibitor using classical HTS and SAR, was called a “magical bullet” as it revolutionized the treatment of CML. Imatinib soon faced resistance due to mutation in Bcr-Abl. Nilotinib was then derived from imatinib using multiple co-crystal structures of Abl [72] and is now used to treat imatinib-resistant CML. X-ray crystallography, together with NMR and hydrogen-deuterium exchange mass spectrometry, was also instrumental for the characterization of allosteric inhibitors of Bcr-Abl [73].

The first example of successful FBDD is vemurafenib (also known as ZELBORAF or PLX4032, Plexxikon/Roche), which inhibits the serine-threonine kinase BRAF V600E mutant found in about half of all melanomas (Table 1). Here, scaffold-based drug discovery was used to identify novel kinase inhibitors starting from small molecules with molecular mass in the range 150–350 Da [74,75]. Initial screening was performed using a biochemical assay against five different kinases and a library containing 20,000 compounds, 238 of which were selected for crystallographic analysis leading to 100 structures of complexes with kinase domains. This provided the basis for iterative, structure-based, rational synthesis where 100 different compounds could be co-crystallized with a highly soluble form of the BRAF kinase domain, finally leading after iterations to the optimized vemurafenib [76]. The use of a unique co-crystal structure has also been a key factor to guide the rational design of venetoclax (also known as VENCLEXTA or ABT-199; AbbVie/Genentech), a first-in-class BCL-2–selective inhibitor [77]. The BCL-2 (B cell chronic lymphocytic leukemia/lymphoma 2) family of proteins comprises key regulators of the apoptotic process. This family includes proapoptotic and prosurvival proteins, and it is well established that shifting the balance toward the latter provides a mechanism whereby cancer cells evade apoptosis. Thus, these proteins represent validated, high-value cancer targets. Inhibiting BCL-2 represents in fact the second success story of FBDD. Here, however, NMR-based screening combined to parallel synthesis was used in the early steps of drug development [78]. This allowed the development of the potent navitoclax BCL-2 inhibitor [79], which could not be used in effective treatment because it induced rapid, concentration-dependent, decreases in circulating platelets. Re-engineering navitoclax led to venetoclax, which also inhibits BCL-2 but spares human platelets [80]. The third fragment-derived drug, erdafitinib (also known as BALVERSA, Janssen Pharmaceutical Companies), was approved by the FDA in April 2019 [81]. This compound was identified based on screening of the Astex proprietary library followed by rationally designed chemical modifications [82]. Erdafitinib has potent tyrosine kinase inhibitory activity against all four members of the fibroblast growth factor receptor (FGFR) family with genetic alterations, the latter being associated with increased tumor growth, metastasis, and angiogenesis. It is prescribed to treat adults with locally advanced and unresectable or metastatic bladder cancer (urothelial carcinoma) with FGFR alterations [83].

## 7. Discussion-Comparison with other Methods: serial Crystallography, cryo-EM, micro-ED, NMR, Neutron Crystallography

Over the past ten years, we have witnessed the development of X-ray-free electron lasers (XFELs) and their applications in many scientific fields, primarily protein crystallography [84]. The peak brightness obtained with such sources is 10 billion times higher than that obtained with third generation synchrotrons. However, the femtosecond X-ray pulses delivered by XFELs are so short that “diffraction-before-destruction” is observed since the damage will occur after the diffracted radiation has left the sample [85]. This led to the development of serial femtosecond crystallography (SFX) where tiny (i.e., micrometer- to nanometer-sized crystals) are used for data collection [86,87,88]. SFX proved early on to be very powerful in unveiling very difficult-to-solve structures, including those of membrane proteins. Pioneering examples are the XFEL structures of the *Blastochloris viridis* photosynthetic reaction center (total structure weight of all non-water atoms in the asymmetric unit: 146 kDa, 3.5 Å) [89] and those of the human serotonin 5-HT_2B_ G protein-coupled receptor (53 kDa, 2.8 Å) [90,91] and *Thermosynechococcus vulcanus* photosystem II (730 kDa, 1.95 Å) [92]. It allowed also the in vivo crystal structure determination of the *Bacillus thuringiensis* Cry3A toxin (67 kDa, 2.9 Å) obtained from injecting bacterial cells into an XFEL beam [93]. More recently, SFX methods have been applied to collecting diffraction data on microcrystals with synchrotron radiation leading to serial synchrotron crystallography (SSX) [94]. One of the advantages of SFX and SSX is that they allow crystal structure determination at room temperature, which yields physiologically more relevant structures. An additional advantage of SFX with respect to drug discovery (reviewed in [95]) is that it allows time-resolved measurements and hence the observation of structural changes upon ligand binding, including changes in water structure.

Whereas membrane proteins represent approximately 30% of all known proteins and are over 60% of drug targets, they represent only 2% of existing crystal structures. As discussed in the previous paragraph, this gap might be filled using XFEL radiation but also and especially by cryo-EM [96], whose advantages and limitations with respect to drug discovery have been extensively reviewed by Renaud et al. [97]. One clear advantage of X-ray crystallography in drug discovery still remains its much faster throughput in all respects from selecting crystals to data collection to structure determination, which renders crystallographic ligand screening easy, whereas it would be 2–3 orders of magnitude slower using cryo-EM [97]. Another important point is that, although getting better, resolutions attained by cryo-EM are typically lower than those observed in X-ray crystal structures. Despite the design of ligands using cryo-EM is underdeveloped so far, it has allowed the study of interactions of known drugs with several targets as exemplified by the study of the mode of inhibition of the *Plasmodium falciparum* ribosome by mefloquinine at 3.2 Å [98]. This has to be compared to the first structural data obtained using X-ray crystallography for a complete eukaryotic ribosome, where 16 crystal structures of complexes of the *Saccharomyces cerevisiae* 80S ribosome with 12 eukaryote-specific and 4 broad-spectrum inhibitors were obtained at high resolution, up to 2.9 Å [99].

Another very promising technique for structural determination is the so-called microcrystal electron diffraction (MicroED), which can be viewed as a cryo-EM technique that uses 3D nano- or microcrystalline arrays (reviewed in [100]). Since its development in 2013 [101] and further improvement due the introduction of continuous-rotation data collection as in X-ray crystallography [102], micro-ED has been used to determine the structures of proteins, oligopeptides, and organic molecules [100]. The potential of this technique for the pharmaceutical industry has been illustrated in two recent works. First, it was used to characterize the structure of active pharmaceutical ingredients in the form of submicrocrystalline powders isolated from drug pills [103]. Second, its application to structure-based drug design was illustrated by the characterization of the binding of bevirimat, a HIV maturation inhibitor, to a construct comprising the C-terminal domain of the CA capsid protein and the SP1 peptide at 2.9 Å resolution [104]. The study of microcrystalline samples using the X-ray powder diffraction (XRPD) technique, in addition to its use for the above-mentioned characterization of drugs, may also be used to study proteins, in particular for preliminary structural characterization [105].

Since the first NMR-determined protein structure in 1985 [106], NMR has become a major contributor in structural biology. Protein-observed NMR requires labelling (^15^N, ^13^C) and is generally limited to proteins of size smaller than about 50 kDa, as peak width grows with protein size, because of faster transversal relaxation, resulting in overlapping signals. The TROSY (transverse relaxation-optimized spectroscopy) experiment relieves this limitation and allows the study of systems with masses close to 1 MDa [107]; however, not to the point that an atomic model can be derived. Besides the use of NMR as a target structure determination tool, which can nicely overcome limitation of X-ray crystallography for partially intrinsically disordered proteins or proteins reluctant to crystallize, NMR excels at the detection and the characterization of intermolecular interactions. Whereas the determination of the crystallographic structure of a protein-ligand complex implies that about 60–70% of the protein in the crystals are in the bound state, which requires high concentrations of ligand when the affinity is low, NMR is able to detect binding when as low as a few percent of the proteins are in the bound state [108]. Another strength of ligand-observed NMR is that all measures are usually performed in solution with chemical label-free proteins and ligand. This allows for large conformational changes upon interaction, which are prevented by crystal lattice interactions in the case of crystals soaking experiments, and also ensures that the ligand-binding site is always accessible. Ligand-observed NMR can serve as a primary screening tool in order to sort out binders from non-binders. This is especially suited to FBLD, where cocktails of fragments can be screened, or as a validation method of primary hits identified by HTS or other biophysical methods. Actually, the FBLD concept emerged from the pioneering work of Shuker and collaborators [109]. Ligand-observed NMR relies on the difference in the NMR characteristics of a ligand between the free and the bound states [108]. Relaxation rates depending on rotational diffusion and the nuclear overhauser effect (NOE), at the root of the most popular experiments STD (saturation-transfer difference) and WaterLOGSY (water-ligand observed via gradient spectroscopy) [108] are the more frequently used. Beyond ligand identification, protein-observed NMR, which requires at least ^15^N-labeling for protein of less than 30 kDa and ^13^C-methyl labelling for larger proteins with a limit of about 100 kDa, can provide valuable information about the ligand-binding site, thanks to chemical shift perturbation experiments, at the condition that the protein resonance has previously been assigned.

One limitation of macromolecular X-ray crystallography is its inability to experimentally observe hydrogen atoms, unless ultra-high-resolution data are available, better than 1.0 Å. Even in that case, protonation states of ionizable groups are not always observable. In macromolecular X-ray crystallography, hydrogen atom positions are usually derived from the observed positions of heavier atoms (C, N, O, and S), on the analysis of possible hydrogen-bond networks and on the knowledge of theoretical pK_a_ of ionizable groups. Yet, the role of hydrogen atoms in ligand recognition and binding by proteins is of paramount importance, as binding occurs through directional hydrogen bonds and depends on the protonation state of ionizable groups in both ligand and protein. Neutron protein crystallography nicely complements X-ray crystallography in that respect [110], especially when deuterated samples are used, as the coherent scattering length of deuterium is of similar amplitude to those of carbon, nitrogen, and oxygen atoms. Partial sample deuteration is performed by the use of D_2_O in the crystallization process, whereas sample per-deuteration can only be achieved during protein expression and chemical synthesis. However, because of technical limitations, neutron crystallography cannot easily be used as a routine tool: as data collection relies on neutron sources with a limited flux, it typically takes days to weeks. Neutron crystallography recently resulted in an in-depth analysis of human carbonic anhydrase in complex with clinical drugs [111] that allowed to determine the ionization state of the bound drugs. In another study, the protonation state of all histidine residues of galectin-3 in interaction with lactose or glycerol could be elucidated [112]. As all potent inhibitors of galectin-3 have been derived from natural ligands, such knowledge is an invaluable help to guide further drug development.

## 8. Conclusions

Crystallography, a multidisciplinary science par excellence—at the confluence of physics, chemistry, mathematics, biophysics, biology, medicine, and earth sciences—has been instrumental in a very large part of the scientific studies of matter for more than a century. This has resulted in an impressive number of Nobel Prize laureates—mostly in physics or chemistry—related to crystallography. Indeed, twenty-eight Nobel Prizes have been awarded either for the founding works of the discipline, such as those of Röntgen, von Laue, and Bragg at the beginning of the 20^th^ century, or for essential contributions to understanding of living matter. In the last decade, crystallography has become, together with other methods such as NMR, cryo-EM, or mass spectrometry, a major player in integrative structural biology for the multi-scale analysis of living organisms and in chemical biology for the understanding of biological problems from a mechanistic point of view. Thanks to ongoing upgrades on synchrotrons, we can objectively believe that macromolecular crystallography will continue to play a major role in life sciences in the future [113]. It has also played and continues to play a key role in drug discovery, particularly in helping to improve molecules in medicinal chemistry programs. However, thanks to the technological innovations that have marked its development since the 1990s, notably automation in the post-genomic era, crystallography has been able to elevate its status from a hit2lead aid to a fast, accurate, and efficient method for identifying hits [114].

## Figures and Tables

**Figure 1 molecules-25-01030-f001:**
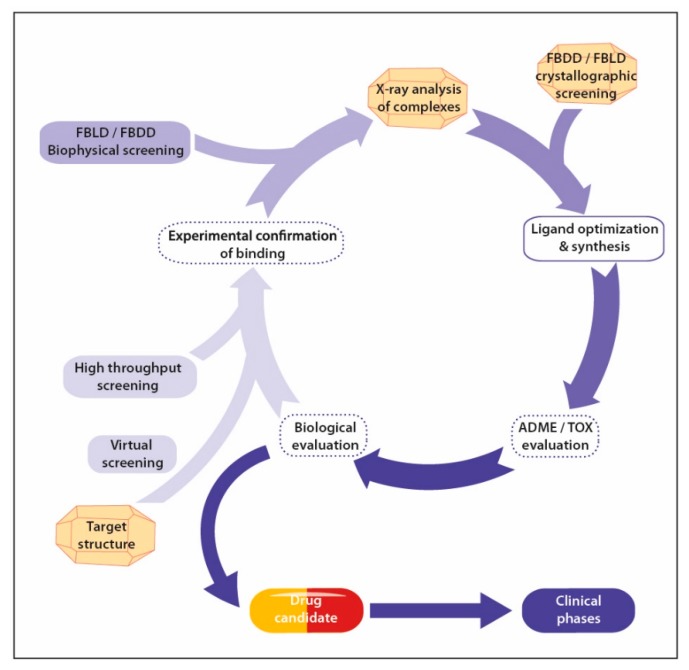
The drug design cycle. Steps in dashed boxes are not mandatory in the early stage of drug development. Contributions of X-ray crystallography are indicated with schematic crystals. Abbreviations: FBLD, fragment-based ligand/lead discovery; FBDD, fragment-based drug discovery.

**Figure 2 molecules-25-01030-f002:**
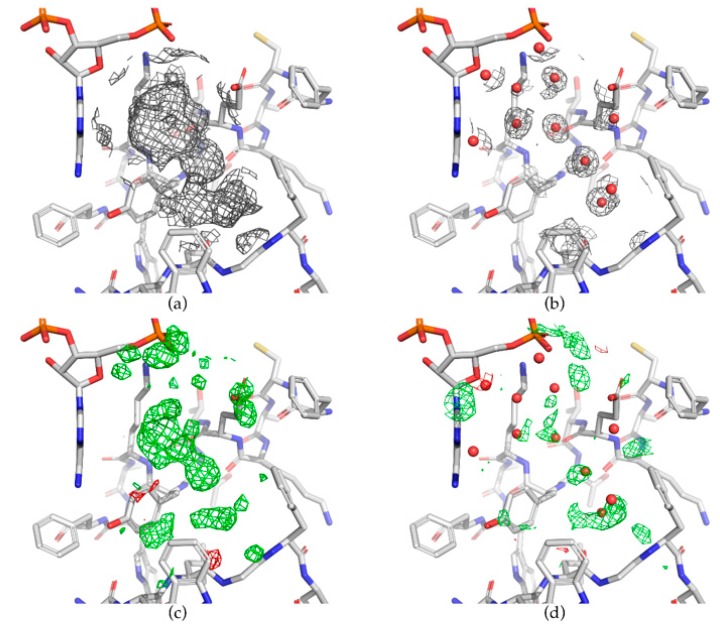
Weakly bound fragments can easily be overlooked. Comparison of maps obtained using the PanDDA procedure (**a**, **c**) or after standard refinement (**b**, **d**). Diffraction data were collected from 839 crystals of an in-house target, each obtained in the presence of a distinct fragment (unpublished results). One specific dataset from this ensemble is shown (resolution of 1.3 Å). Although the water molecules displayed in (**b**) and (**d**) were also included in the model fed to the PanDDA procedure, they were not included in (**a**) and (**c**) to improve clarity. The PanDDA event map (**a**, contoured at 1 σ, BDC = 0.88) and the Z-map (**c**, green/red ±3 contour level) clearly show the presence of the fragment. Standard 2mF_o_-DF_c_ (**b**, 1 σ contour level) and mF_o_-DF_c_ (**d**, green/red ±3 σ) electron density maps, obtained after preliminary refinement (*R*/*R_free_* 0.172/0.193), fail to indicate fragment binding. All maps were carved 5 Å around the position of the bound fragment. For a definition of BDC and Z-map, please refer to [62].

**Table 1 molecules-25-01030-t001:** Examples of fragment-based ligand/lead discovery (FBLD)-derived drugs/compounds (adapted from [35]).

Drug	Company	Target name, Function
	Approved	
Vemurafenib/ZELBORAF	Plexxikon/Roche	BRAF, protein kinase (mutation V600E)
Venetoclax/VENCLEXTA	AbbVie/Genentech	BCL-2, apoptosis suppressor protein
Erdafitinib/BALVERSA	Janssen Pharmaceutical Companies	FGFRs, fibroblast growth factor receptors
	**Phase III**	
Asciminib/ABL001	Novartis	Bcr-Abl, oncoprotein
Lanabecestat ^1^	AstraZeneca/Eli Lilly and Company	BACE-1, β-site amyloid precursor protein-cleaving enzyme 1
Pexidartinib/TURALIO^TM^	Plexxikon/Daiichi Sankyo	CSF1R, colony stimulating factor 1 receptor KIT, proto-oncogene receptor tyrosine kinase
Verubecestat ^1^	Merck	BACE-1, β-site amyloid precursor protein-cleaving enzyme 1
	**Phase II**	
AT7519 ^2^	Astex	CDKs, cyclin-dependent kinases
AT9283	Astex	Aurora and JAK2, kinases
Luminespib/AUY-922 ^1^	Vernalis/Novartis	HSP90, heat shock protein
Capivasertib/AZD5363	AstraZeneca/Astex	AKT, serine/threonine protein kinase
CPI-0610	Constellation	BET, bromodomain and extra-terminal protein
DG-051 ^2^	deCODE	LTA4H, leukotriene A4 hydrolase
eFT508	eFFECTOR	MNK1/2, kinases
Indeglitazar ^2^	Plexxikon	PPARs, peroxisome proliferator-activated receptors
LY2886721 ^2^	Eli Lilly and Company	BACE-1, β-site amyloid precursor protein-cleaving enzyme 1
LY517717 ^2^	Eli Lilly and Company	FXa, Factor Xa
Navitoclax/ABT-263	Abbott	BCL-2/BCL-XL, apoptosis suppressor proteins
Onalespib/AT13387	Astex	HSP90, heat shock protein
PF-06650833	Pfizer	IRAK4, interleukin-1 receptor-associated kinase
PF-06835919	Pfizer	KHK, ketohexokinase
	**Phase I** ^3^	

^1^ Compounds for which the corresponding clinical phase has failed or has been discontinued; ^2^ Compounds whose development status is unknown; ^3^ In October 2018, at least nineteen fragment-based ligand/lead discovery (FBLD)-derived drugs were in Phase I.

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
