# Peer review of "Protein X-ray Crystallography and Drug Discovery"

_molecules, 2020, doi:10.3390/molecules25051030_

Round 1

Reviewer 1 Report

The review article by Maveyraud and Mourey is a comprehensive story about using protein x-ray crystallography in drug discovery. I read it with real pleasure and I would like to express my gratitude to the authors, it is a really valuable piece. I highly-recommend publishing it in Molecules journal and I am convinced it will be appreciated by the scientific community. I also recommend small corrections:

Line 20 - "X-crystallography" change to "X-ray crystallography"

Line 96 - for SIR/MIR one may mention other elements e.g. Hg, 

You may also introduce S-SAD (phasing without Se and heavy atoms)

Line 158 - I just read a very comprehensive review by Kirsch et al Molecules 201924(23), 4309; it is worth to direct readers to this work

Line 189 - seeding not described

line 209 - high-molecular-weight 

Line 241 - it surprises me that MASSIVE has an average throughput only 120 xtals a day. Especially using such brilliant source. 

General: The Authors may consider crating Boxes with Glossaries of used terms (occupancy, ADP, MR, SAD, MolRep, etc.). It has been utilized e.g. in Nature Reviews and it really helps the non-specialist reader. 

Author Response

- Line 20 - "X-crystallography" change to "X-ray crystallography"

Manuscript has been modified accordingly.

- Line 96 - for SIR/MIR one may mention other elements e.g. Hg

Manuscript has been modified accordingly.

- You may also introduce S-SAD (phasing without Se and heavy atoms)

S-SAD phasing method has been introduced, with a proper reference (reference N° 19 in the revised version).

- Line 158 - I just read a very comprehensive review by Kirsch et al Molecules 2019, 24(23), 4309; it is worth to direct readers to this work.

The corresponding reference has been added as reference N° 26 in the revised version.

- Line 189 - seeding not described.

A short description of seeding has been introduced.

- line 209 - high-molecular-weight 

Manuscript has been modified accordingly.

- Line 241 - it surprises me that MASSIVE has an average throughput only 120 xtals a day. Especially using such brilliant source. 

This is the average throughput reported in the paper cited as a reference (Hutin et al., 2019, J. Vis. Exp., 145:e59032 DOI: doi:10.3791/59032) written by the people  running the beam line. This agrees well with our own experience: we collected about 840 crystal within 7 days of beamtime. 

- General: The Authors may consider crating Boxes with Glossaries of used terms (occupancy, ADP, MR, SAD, MolRep, etc.). It has been utilized e.g. in Nature Reviews and it really helps the non-specialist reader. 

The authors thank the reviewer for this suggestion. However, most of the terms used and all abbreviations are defined in the text. In this context, it does not seem useful to add boxes.

Reviewer 2 Report

The review paper by Maveyraud and Mourey covers a very important topic in the field of X-ray crystallography.

The presentation is very clear and the discussed references are more than adequate and recent.

I therefore support its pubblication in "Molecules" in the present form.

Author Response

N/A

Reviewer 3 Report

The authors report in this review work on the X-ray crystallography as a characterization technique very useful in the drug discovery process involving proteins. The work covers several sections, from theoretical and practical aspects to some reported examples of discoveries and developments associated to drugs and proteins. Finally, it is reported a comparison with other characterization techniques as, for instance, NMR. Overall, the work is well-researched, well-organized and well-written, but not at all well-illustrated. Given that the authors are dealing with a review work, the only two reported figures (Figures 1 and 2) are not enough to cover the whole study. In my opinion, more figures that illustrate the reported examples and discussions should be added.

Other points to address are:

In page 3 (line 79): the authors should indicate suitable references at the end of the sentence: “…reviews and textbooks are available [references?].” In caption of Table 1. Please, remove from the caption the web address http://.... It could be transferred to the references section. In page 13 (line 471): correct the angstrom symbol Å at the end of the sentence “…better than 1.0 A.” Please, check out the journal name abbreviation in the references section and write it in the correct form, see references No. 5, 6, 9, 15, 16, 27, 55, 86, 88, 90, 92, 104.

My recommendation is to publish the work in the journal Molecules, after addressing the indicated revisions.

Author Response

- Overall, the work is well-researched, well-organized and well-written, but not at all well-illustrated. Given that the authors are dealing with a review work, the only two reported figures (Figures 1 and 2) are not enough to cover the whole study. In my opinion, more figures that illustrate the reported examples and discussions should be added.

Two figures and a table seem reasonable for a review. Moreover, the figures chosen (one of which is based on unpublished results) illustrate particular points, leaving interested readers to resort to the numerous, mostly recent, referenced articles.

- In page 3 (line 79): the authors should indicate suitable references at the end of the sentence: “…reviews and textbooks are available [references?].”

A reference (reference N° 18 in the revised version) to a very recently published book chapter has been added.

- In caption of Table 1. Please, remove from the caption the web address http://.... It could be transferred to the references section.

Manuscript has been modified accordingly, i.e. the web address has been transferred to the "References" section (it now appears as reference N° 36 in the revised version).

- In page 13 (line 471): correct the angstrom symbol Å at the end of the sentence “…better than 1.0 A.”

Manuscript has been modified accordingly.

- Please, check out the journal name abbreviation in the references section and write it in the correct form, see references No. 5, 6, 9, 15, 16, 27, 55, 86, 88, 90, 92, 104.

All references have been checked and modified when necessary.

Reviewer 4 Report

This is a brilliant review article on "Protein X-ray Crystallography and Drug Discovery". Very nicely written and presented including quite a lot of prior knowledge. Our only recommendation would be to include one more method which has been developed exactly for the purpose of crystal and drug screening using microcrystalline protein precipitates in the absence of good quality single crystals. The latter is the X-ray Powder Diffraction technique. Especially in the section "7. Discussion - Comparison with other methods: serial crystallography, cryo-EM, micro-ED, NMR, neutron crystallography", we would recommend the method to be briefly described together with the rest of the techniques. We recommend the following review articles and Book chapters to be potentially cited in this very nice review-

2. “Macromolecular Powder Diffraction: Ready for Genuine Biological Problems”, Protein & Peptide Letters, 23 (3):232-41 (2016), F. Karavassili & I. Margiolaki

3. "Applications of X-ray Powder Diffraction in Protein Crystallography and Drug Screening ." Crystals 2020, 10, 54, M. Spiliopoulou, A. Valmas, D. P. Triandafillidis, C. Kosinas, A. Fitch, F. Karavassili and I. Margiolaki

4. “Macromolecular Powder Diffraction”, Book Chapter for the International Tables of Crystallography- Volume H: Powder Diffraction, chapter 7.1, 718-736 (2019, available online), I. Margiolaki

Author Response

- This is a brilliant review article on "Protein X-ray Crystallography and Drug Discovery". Very nicely written and presented including quite a lot of prior knowledge. Our only recommendation would be to include one more method which has been developed exactly for the purpose of crystal and drug screening using microcrystalline protein precipitates in the absence of good quality single crystals. The latter is the X-ray Powder Diffraction technique. Especially in the section "7. Discussion - Comparison with other methods: serial crystallography, cryo-EM, micro-ED, NMR, neutron crystallography", we would recommend the method to be briefly described together with the rest of the techniques. We recommend the following review articles and Book chapters to be potentially cited in this very nice review-

2. “Macromolecular Powder Diffraction: Ready for Genuine Biological Problems”, Protein & Peptide Letters, 23 (3):232-41 (2016), F. Karavassili & I. Margiolaki

3. "Applications of X-ray Powder Diffraction in Protein Crystallography and Drug Screening ." Crystals 2020, 10, 54, M. Spiliopoulou, A. Valmas, D. P. Triandafillidis, C. Kosinas, A. Fitch, F. Karavassili and I. Margiolaki

4. “Macromolecular Powder Diffraction”, Book Chapter for the International Tables of Crystallography- Volume H: Powder Diffraction, chapter 7.1, 718-736 (2019, available online), I. Margiolaki

We have added a sentence in the chapter "Discussion - Comparison with other methods: serial crystallography, cryo-EM, micro-ED, NMR, neutron crystallography" and one of the proposed references (reference N° 106 in the revised version).